# Cellular Reprogramming—A Model for Melanoma Cellular Plasticity

**DOI:** 10.3390/ijms21218274

**Published:** 2020-11-05

**Authors:** Karol Granados, Juliane Poelchen, Daniel Novak, Jochen Utikal

**Affiliations:** 1Skin Cancer Unit, German Cancer Research Center (DKFZ), D-69120 Heidelberg, Germany; karolandrea.granados@ucr.ac.cr (K.G.); j.poelchen@dkfz-heidelberg.de (J.P.); d.novak@dkfz-heidelberg.de (D.N.); 2Department of Dermatology, Venereology and Allergology, University Medical Center Mannheim, Ruprecht-Karl University of Heidelberg, D-68135 Mannheim, Germany; 3Department of Biochemistry, School of Medicine, University of Costa Rica (UCR), Rodrigo Facio Campus, San Pedro Montes Oca, San Jose 2060, Costa Rica

**Keywords:** melanoma, cellular plasticity, heterogeneity, partial reprogramming, phenotype switch

## Abstract

Cellular plasticity of cancer cells is often associated with phenotypic heterogeneity and drug resistance and thus remains a major challenge for the treatment of melanoma and other types of cancer. Melanoma cells have the capacity to switch their phenotype during tumor progression, from a proliferative and differentiated phenotype to a more invasive and dedifferentiated phenotype. However, the molecular mechanisms driving this phenotype switch are not yet fully understood. Considering that cellular heterogeneity within the tumor contributes to the high plasticity typically observed in melanoma, it is crucial to generate suitable models to investigate this phenomenon in detail. Here, we discuss the use of complete and partial reprogramming into induced pluripotent cancer (iPC) cells as a tool to obtain new insights into melanoma cellular plasticity. We consider this a relevant topic due to the high plasticity of melanoma cells and its association with a strong resistance to standard anticancer treatments.

## 1. Introduction

Cellular plasticity has been widely studied during normal development, cell fate determination, wound repair, and more recently as a hallmark of cancer cells promoting dissemination and resistance to treatments [1,2,3]. The occurrence of different phenotypic states during cancer progression involves a process of dedifferentiation. Dedifferentiation describes the phenotypic and molecular transformation of a terminally differentiated or committed cell towards a less differentiated stage with greater developmental potential, that is sustained by specific gene expression, epigenetic events, environmental factors, and connected to other cellular processes [1,4].

Melanoma cells have the capacity to adopt different cellular phenotypes during tumor growth and metastasis which is one reason for melanoma being a highly aggressive type of cancer [5]. The ability of melanoma cells to adapt to environmental changes and to undergo phenotype switch during tumor progression, it provides a suitable platform to study dedifferentiation and obtain new insights into cellular plasticity of cancer cells. 

According to Waddington’s epigenetic landscape model, cells develop from a progenitor stem cell to a mature, fully differentiated and functional cell. In this model, cells are depicted as balls and distinct differentiation routes are represented as hills and valleys. Like the ball rolling down from a hill into deeper valleys, a less differentiated cell, e.g., a stem cell, transforms into more differentiated cells. The lowest valley is reached when a cell is terminally differentiated and has no further differentiation potential [6]. Although this process was thought to be unidirectional, research on different cellular phenomena such as dedifferentiation, transdifferentiation and reprogramming has shown that the process can be reversed [7] (Figure 1).

The process of transforming a terminally differentiated cell into a more primordial one is called dedifferentiation and the complete dedifferentiation towards a pluripotent stage is known as cellular reprogramming [1]. Although reprogramming involves dedifferentiation as part of the process, it is important to clarify that reprogramming means complete reversion towards pluripotency, while dedifferentiation rather describes a reversible step backwards towards a less differentiated progenitor of the original cell [1].

Considering this, cancer progression and cellular reprogramming share many common features. Several reports have demonstrated that different types of malignant cells indeed express transcription factors that are usually only found in embryonic stem cells (ESCs). Moreover, they lose their original cell identity and begin to resemble pluripotent-like cells [8,9]. These observations indicate that the development and progression of cancer might rely on molecular mechanisms that also operate the maintenance of cell fate and the self-renewal of progenitor cells.

In many cancer types a specific subpopulation of cells within the tumor possesses the capacity to self-replicate indefinitely, to withstand standard anticancer therapies and to give rise to different cell types that constitute the tumor mass. Due to their stem cell-like properties they are called cancer stem cells (CSCs) [8,10]. The CSC model suggests a hierarchy where a very small population of immortal CSCs at the top generates all the different types of terminally differentiated cells within a tumor, the non-stem cancer cells (NSCCs), which are responsible for the high tumor heterogeneity [8,11]. However, in addition to the hierarchical CSC model, the plastic CSC model suggests a dynamic bidirectional transition between CSCs and differentiated cells. According to this, differentiated cells can acquire CSC-like features and dedifferentiate into stem-like cells under certain conditions causing an increment in tumor cell plasticity [9,11,12,13]. Several studies report the bidirectional interconversion for different mammalian organ systems like the lung, intestine, heart and breast [14,15,16,17,18] (Figure 1). 

Deciphering what exactly happens on a molecular level during cellular reprogramming might also help us to understand cancer initiation, progression and recurrence in a way that will enable us to improve the efficacy of treatments considerably. For this purpose, new in vitro models for studying cellular plasticity and CSCs are required [19]. In this review, we summarize the current knowledge about cellular reprogramming and the use of partial reprogramming as a model to study cellular plasticity during melanoma progression.

## 2. Cellular Reprogramming and iPS Cells

Based on their differentiation potential, cells can be classified as totipotent, pluripotent, multipotent or unipotent [20,21]. Pluripotent cells include ESC, and induced pluripotent stem (iPS) cells. 

All pluripotent cells have the potential to differentiate into every ecto-, meso- or endodermal cell type of the body [22,23]. Indeed, ESC and iPS cells both can form encapsulated tumors called teratomas that consist of differentiated cells from all three embryonic germ layers [20,21,22]. 

To the present day, a wide range of different somatic cells has been converted to iPS cells by a method termed cellular reprogramming. Successful reprogramming requires genomic manipulation, changes in the cellular environment or both. By performing some groundbreaking experiments in 1958, John Gurdon became a pioneer in the field of cellular reprogramming. He and his collaborators demonstrated that the transfer of a nucleus from cells of different developmental stages from embryonic to somatic cells of pre-hatching Xenopus tadpoles into an unfertilized egg induced nuclear reprogramming. After the transfer, the nuclei were able to initiate normal development of tadpole embryos right up to normal adult frogs [24]. 

More recently, iPS cells have been successfully generated from mouse and human fibroblasts by direct reprogramming using a combination of four transcription factors (Yamanaka factors, OSKM): Octamer-binding transcription factor 3/4 (Oct4), Sex-determining region Y-box 2 (Sox2), Kruppel-like factor 4 (Klf4) and c-Myc [25,26] that were ectopically expressed in somatic cells. Since the first successful reprogramming with the Yamanaka factors, different combinations of transcription factors, including Nanog and Lin28 have been used to successfully produce iPS cells in vitro and in vivo from somatic cells of different organisms [27,28,29]. Other strategies for cellular reprogramming include cell fusion [30], transfection with microRNAs [31], and even the use of small molecules that enhance efficiency of cellular reprogramming [32], including vitamin C [33], valproic acid (histone deacetylase inhibitor) [34] or 5-azacytidine (DNA methyltransferase inhibitor) [35].

Reprogramming of somatic cells is a complicated multi-step process that involves different phases (early, intermediate and late phase) [36]. During the early/initiation phase, induction of mesenchymal-to-epithelial transition (MET) is a classic event. It goes along with a downregulation of lineage-specific genes, an increased proliferation and also metabolic changes [37]. The following intermediate phase includes stochastic activation of pluripotency markers and transcription factors that regulate genes normally expressed during early development/differentiation in pluripotent stem cells. Early and intermediate phases are both regulated by ectopic OSKM expression and during these phases the cells are considered to be only partially reprogrammed, because they are still in an epigenetically unstable cell state [36]. Partial reprogramming will be discussed in detail later in this review. Finally, during the late/maturation phase, the core pluripotency transcriptional circuitry is activated and renders the cells independent of the ectopic expression of the pluripotency factors, and the epigenetic marks are globally reset [37].

Although cellular reprogramming seems to be a stochastic process with randomly activated and repressed genes, it has been shown that specific epigenetic changes need to happen (e.g., DNA methylation and histone modifications) and specific genetic programs need to be activated in order to enable complete reprogramming [38]. In the initial stage of reprogramming, the ectopic expression of the transcription factors Oct4, Sox2 and Klf4 induces the endogenous expression of factors that drive the transition to pluripotency, such as ESRRB and Sall4. Among the reprogramming factors Oct4 is of particular importance as it initiates the expression of the genes Nanog, POU5F1 and MYOD1 [36]. Moreover, c-Myc and either of the other factors can bind simultaneously to specific genetic elements, resulting in chromatin decondensation followed by enhanced gene expression [39]. The reprogramming transcription factors also induce a switch in the metabolic network of the cells, which enhances the efficiency of cellular reprogramming [40,41].

Successful and complete reprogramming requires the stable expression of endogenous stemness-related genes independent of exogenous expression of reprogramming transcription factors. Oct4 and Nanog are important genes playing important roles in embryogenesis and it is not surprising that, together with SOX2, they are most important for reprogramming [42]. These three transcription factors form the so-called core transcriptional regulatory circuitry by regulating and maintaining their own expression as well as the expression of pluripotency-related genes. In addition to the activation of pluripotency-associated genes, the maintenance of the pluripotent state also depends on the suppression of lineage-specifying genes [42]. Interestingly, pluripotent and adult stem cells as well as cancer cells share these characteristics described above.

## 3. Reprogramming of Cancer Cells

Dedifferentiation as well as cellular reprogramming both go along with lots of transcriptional and epigenetic changes. For this reason, cellular reprogramming could be a suitable tool for disease modeling, including the modelling of cancer progression [18,43,44]. Dedifferentiation constitutes an undeniable characteristic of cancer progression due to its contribution to cancer cell plasticity and therapy resistance [2,12]. Certainly, the functional and phenotypic heterogeneity among the cells constituting the tumor mass can be explained by reprogramming processes [45]. It is well known that transcription factors like Oct4, Sox2, Nanog, Klf4 and Lin28 which facilitate cellular reprogramming, also promote tumorigenesis and tumor progression [8,9]. Moreover, it has been reported that cells within poorly differentiated tumors show an embryonic stem cell-like transcriptional profile with stem cell-specific genes being expressed [9,12].

For many cancers, it is believed that genetic and epigenetic changes that occur spontaneously or that are induced by cancerogens induce dedifferentiation and cellular reprogramming of normal stem cells transforming them into CSCs [45] (Figure 2). There is abundant evidence that CSCs contribute to metastasis, drug resistance, and disease relapse. Due to their high telomerase activity and their ability to self-renew indefinitely CSC populations facilitate long-term tumor growth. Additionally, genes and pathways that confer multidrug resistance are highly active in CSCs [46,47]. Currently, several studies address the question how to selectively target and eliminate cancer-initiating cells or CSCs in order to prevent the development of drug resistances and tumor relapse upon therapy.

To investigate dedifferentiation of cancer cells and generation of CSC-like cells, different in vitro and in vivo protocols for the production of iPS cells have been implemented in several types of cancer [48]. Dedifferentiation by ectopic expression of reprogramming transcription factors such as OSKM, have been reported in a variety of cancers including melanoma [27,49,50], Leukemia [51], gastrointestinal cancer [52], osteosarcoma [53], colorectal cancer [52], hepatocellular carcinoma [54], bladder cancer [55], neuroblastoma [56], pancreatic ductal adenocarcinoma [57] and others. However, the molecular mechanisms that underlie the process of dedifferentiation are still not completely understood [19,58]. 

It is known that as a consequence of reprogramming cancer cells towards induced pluripotent cancer (iPC) cells, the tumorigenic potential of iPC cells decreases while the potential to differentiate into other cell types increases [48,50] (Figure 2). An additional advantage of reprogramming cancer cells is the possibility to produce a significant number of CSC-like cells to explore their properties and molecular mechanisms involved in drug response. Therefore, iPC cells have the potential to be used for pharmacological screenings in order to discover new therapeutic targets [59].

Furthermore, it is well accepted that tumor heterogeneity plays a significant role in drug response and minimal residual disease in cancer [60,61]. In melanoma, tumor heterogeneity enables the coexistence of multiple melanoma cell phenotypes that respond differently to standard cancer therapies [62]. Castro-Perez and collaborators (2019) reported that melanoma-derived iPC cells exhibit an increased mitogen-activated protein kinase (MAPK) inhibitor resistance, suggesting that cellular reprogramming of cancer cells can serve as a model to understand melanoma cell plasticity-dependent mechanisms especially in the recurrence of aggressive drug-resistant melanoma [63]. Further studies on melanoma plasticity could provide a basis for the identification of key factors involved in drug response and resistance.

## 4. Melanoma Plasticity 

Genetic aberrations, influences from the tumor microenvironment and epigenetic changes all contribute to the phenotypic plasticity and high heterogeneity that is characteristic for melanoma [64,65]. Melanoma cells can reversibly alter between a proliferative/differentiated and an invasive/dedifferentiated phenotype, a process comparable to epithelial-to-mesenchymal transition (EMT). This transition of melanoma cells towards an invasive phenotype facilitates tumor dissemination from a primary tumor to distant sites during metastasis progression [66,67,68] (Figure 3). The phenotype alteration can also be induced by a hypoxic tumor microenvironment and inflammatory signals [64,69,70]. The high melanoma heterogeneity and intrinsic plasticity of melanoma cells suggest the presence of both proliferative and invasive phenotypes [5]. 

Among the cellular events that take place during the switch from the proliferative to the invasive phenotype, changes in cadherin expression have been associated with an increment in the invasive capacity of BRAF-mutant melanoma cells [71]. During metastasis, E-cadherin is downregulated while N-cadherin expression levels increase progressively. These expression changes contribute to migration and survival of melanoma cells [64,67,71]. Furthermore, Bettum and colleagues (2015) demonstrated that the adoption of an invasive phenotype is associated with dedifferentiation and metabolic reprogramming from mitochondrial oxidation to glycolysis, which promotes growth and survival during phenotypic transition [64].

In order to sustain the increase in cell growth and proliferation, cancer cells require a higher energy demand that is mainly supported by aerobic glycolysis (the Warburg effect) under low or even normal levels of oxygen [72,73,74]. BRAF-mutant melanoma cells usually express high level of the hypoxia inducible factor (HIF) that facilitates hyperactivation of MAPK signaling and with that an increase in the proliferation rate [74,75]. On the other hand, when cancer cells switch into an invasive and highly treatment resistant phenotype, mitochondrial oxidative phosphorylation (OxPhos) metabolism is usually induced in slow-cycling cells [75]. This metabolic switch is mediated by different factors such as microphthalmia-associated transcription factor (MITF), histone lysine demethylase called JARID1B, and the peroxisome proliferator-activated receptor gamma coactivator 1 alpha (PGC1alpha) [73,75]. MITF plays a pivotal role in melanoma phenotype switching because it regulates the expression of genes involved in melanocyte differentiation, pigmentation and the dynamic change between the different phenotypic states [76]. MITF is also considered as a lineage-specific oncogene highly expressed in human melanomas that contributes to tumorigenesis [77,78], by promoting reversible and functional reprogramming of signaling pathways in melanoma [79,80]. In response to extracellular signals and environmental factors like UV light, MITF induces the expression of the differentiation-associated pigmentation machinery which includes melanosomal factors and enzymes that are necessary for pigment production [81]. Goding and collaborators (2011) proposed that the MITF activity correlates with the phenotype switching in melanoma tumor cells. According to the “MITF rheostat” model, different expression levels of MITF are connected with specific phenotypic states of melanoma cells. A high level of MITF activity promotes differentiation, a mid-level activity promotes proliferation, a low-level activity promotes an invasive stem cell-like phenotype, and the absence of MITF activity causes senescence or cell death [5,82] (Figure 3). 

A downregulation of MITF in melanoma cells has been related to an invasive/poorly proliferative phenotype, increased plasticity and acquired therapy resistance to BRAF inhibition [83,84,85]. Induction of phenotype switching towards a more dedifferentiated state drastically enhances the aggressiveness of the tumor, and constitutes an essential mechanism underlying therapy resistance in melanoma patients [2,86,87]. Furthermore, deregulation of MITF expression is linked to a reduced efficacy of BRAF and MEK inhibitors, and promotes innate resistance by mediating survival signaling and changing metabolism [38,88,89,90,91].

Determining the expression levels of MITF and other factors that promote metastasis and phenotype switching, allows to distinguish between melanoma cells that are sensitive or resistant to BRAF inhibitor. High expression of MITF and other markers of differentiation such as tyrosinase-related protein1 (TYRP1), pre-melanosomal protein (PMEL) and melan-A (MLANA) is distinctive of sensitive melanoma cells. On the other hand, low MITF expression along with high expression of proteins like Wnt5, NFkB and the receptor tyrosine kinase AXL is a feature of resistant cells [78,79,85]. Furthermore, an inversely correlated expression of receptor tyrosine kinase AXL compared to MITF has been reported to drive tumor resistance [5].

The expression ratio of MITF/AXL has been used to describe early resistance in melanoma [85]. MITF^low^/AXL^high^ expression in treatment naïve samples could be related to a resistant and progressive disease [84,88]. Recent single cell analyses by Tsoi et al. showed the presence of four distinct subtypes in melanoma with different transcriptional profiles: undifferentiated/invasive subtype, neural crest-like subtype, transitory subtype, and melanocytic/differentiated subtype [2]. This study demonstrates that progressive dedifferentiation of melanoma cells towards a more immature invasive state along with the differentiation trajectory, can be seen as an innate and required resistance mechanism to treatment with kinase inhibitors [2]. However, despite recent studies revealing molecular mechanisms of phenotype switching and cellular plasticity in melanoma the process of reprogramming is still not completely understood. 

## 5. Partial Reprogramming of Melanoma Cells 

A more recent strategy to explore intermediate phases during cell fate reversion is partial cellular reprogramming of cancer cells. This technique is a suitable model to investigate dedifferentiation and early plasticity typically exhibited in melanoma. Partial reprogramming of cancer cells involves technologies that have been already used to produce iPS cells, such as ectopic expression of transcription factors, but just for a specific period of time long enough to reset the epigenetic program that regulates and maintains cell identity (Figure 4).

Using viral delivery systems, murine melanoma cells have been partially reprogrammed for 12 to 20 days [49,83]. These partially reprogrammed cells adopt a less proliferative, more invasive and more dedifferentiated phenotype with a reduced expression of melanocyte-specific markers and an induced expression of stemness-related genes. Moreover, dedifferentiated cells become less sensitive to MAPKi therapy, increasing their survival and expression of pluripotency-related markers [83]. 

Non-viral methods have also been used to partially reprogram murine melanoma cells for 12 to 18 days, using four transcription factors: Oct4, Sox2, Lin28 and Nanog [92]. In this method, Câmara and collaborators (2017) transformed melanoma cells into less aggressive murine melanoma cancer cells. Here, cells show similar morphology to iPS cells and expression of pluripotency markers, but no teratoma formation in vivo. However, subsequent tumors were smaller compared to tumors obtained upon injecting the parental B16F10 cells due to the decrease in proliferative capacity of the partially reprogrammed cells, which reduces aggressiveness of tumors and supports the increment in cancer plasticity and melanoma phenotype switching [92]. 

In general, partial or uncompleted reprogramming generates cancer cell populations comprising cells with diverse phenotypes, which resembles the high heterogeneity underlying cancer plasticity in naturally occurring tumors [3] (Figure 4). An epithelial/mesenchymal hybrid phenotype has been reported for partially reprogrammed colorectal cancer cells [93], based on irregular expression patterns of genes involved in MET/EMT processes (vimentin, E-cadherin, and Snai1) and different epigenetic regulatory mechanisms.

Finally, cancer plasticity remains one of the main reasons for the failure of established cancer treatments. Today’s therapeutic options for the treatment of metastatic melanoma are still not efficient enough to avoid recurrence mostly because of the development of resistances. Combination therapies with BRAF and MEK inhibitors led to tumor regression in more than 90% of patients with metastatic melanoma and significantly improved their progression-free (PFS) and overall survival (OS) [94]. However, an important amount of patients relapses during treatment due to the development of acquired resistance. Moreover, up to 10% of patients show no response to targeted therapy at all, displaying an intrinsic or primary resistance [65]. Considering this scenario, partial reprogramming can provide a better understanding of events that lead to malignancy and plasticity in melanoma, contributing to the discovery of early adaptive mechanisms that allow melanoma cells to survive and become resistant to cancer therapies.

Moreover, using complete or partial reprogramming in combination with multi-omics and single-cell technologies can provide a more comprehensive analysis of the elements that control cellular plasticity in tumors [95,96]. In the future, targeting cellular plasticity represents a promising tool in biomedical research in order to decipher the mechanisms behind therapy resistance in cancer patients and to guide investigations towards an approach of personalized medicine.

## 6. Conclusions 

Melanoma cellular plasticity is a complex phenomenon that is the cause of the high heterogeneity, invasiveness and survival capabilities of melanoma cells. Although significant progress in understanding the mechanisms controlling cancer cellular plasticity has been made recently, the molecular mechanisms underlying melanoma phenotype switching and its contribution to drug response, therapy resistance and disease recurrence still need to be decoded.

In this review, we have discussed the use of complete and partial cellular reprogramming as a model to study melanoma cellular plasticity. The ability of melanoma cells to dedifferentiate or reprogram in response to environmental changes can be explained by their derivation from neural crest cells, a multipotent population that can differentiate into different cell types. Dedifferentiated melanoma cells resemble CSCs in their abilities to contribute to tumor maintenance and therapy resistance. The progressive and reversible process of dedifferentiation of melanoma cells into a stem cell-like invasive phenotype can be considered as an escape mechanism to survive targeted and immune therapies. Moreover, we consider that partial reprogramming can provide a better insight into the initial mechanisms underlying cancer plasticity in melanoma cells.

In the near future, we believe that cellular reprogramming of cancer cells, will broaden our knowledge about cancer plasticity. Comprehensive knowledge about melanoma cellular plasticity would help to predict patients’ responses to targeted or immune therapies and would enable the development of novel and more efficient anticancer therapies.

## Figures and Tables

**Figure 1 ijms-21-08274-f001:**
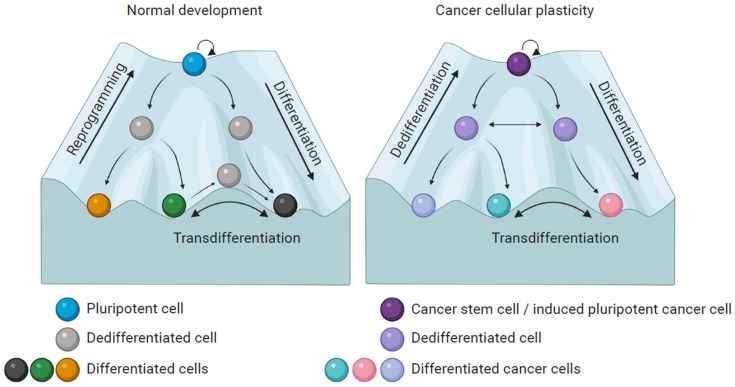
Comparison between normal development and cancer cellular plasticity, based on the model of the epigenetic landscape described by Waddington. Pluripotent cells can differentiate into cells from different lineages. The reverse process from somatic cell to pluripotent cell is known as reprogramming. Additionally, differentiated cells can switch between lineages, a process known as transdifferentiation. In cancer a similar process occurs, in this case, cancer stem cells (CSCs) can generate various differentiated cancer cells and in this way support tumor growth. Moreover, differentiated cancer cells can undergo a reversible dedifferentiation. CSCs, dedifferentiated and differentiated cancer cells together support tumor heterogeneity. Additionally, cancer cells can be artificially converted into induced pluripotent cancer cells by nuclear reprogramming.

**Figure 2 ijms-21-08274-f002:**
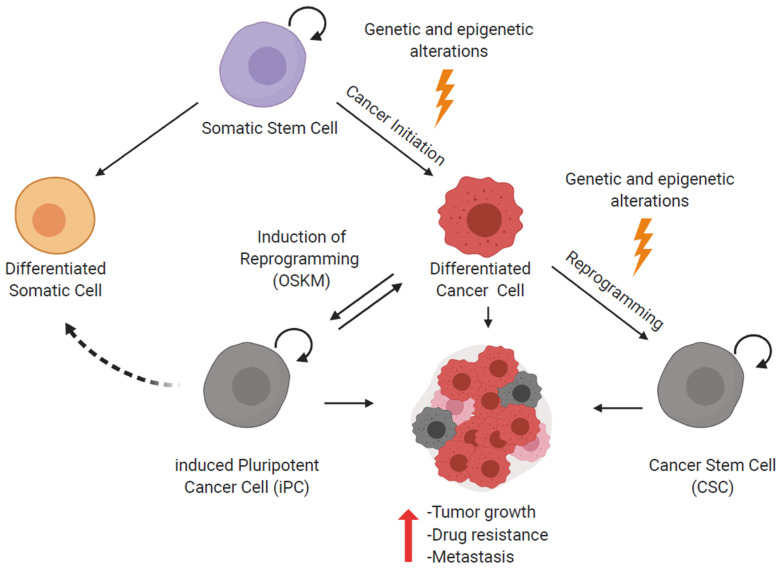
Generation of CSC and reprogramming of cancer cells. Genetic and epigenetic changes can induce the malignant transformation of a somatic stem cell right up to a CSC state. CSCs support tumor growth, drug resistance and metastasis. Moreover, differentiated cancer cells can be reprogrammed towards iPC cells, which show features of CSCs. However, iPC cells can also differentiate into non-tumorigenic somatic cells or tumorigenic, differentiated cancer cells.

**Figure 3 ijms-21-08274-f003:**
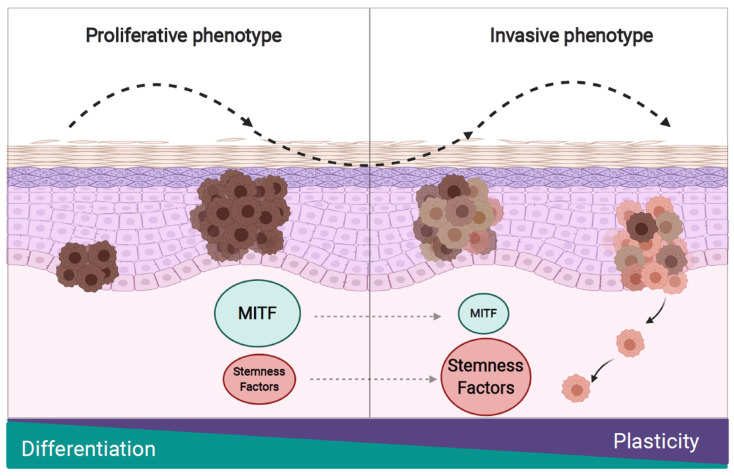
Cellular plasticity in melanoma. Melanoma cells can switch between a differentiated/proliferative and a dedifferentiated/invasive phenotype. This phenotype switch increases plasticity and is responsible for a poor response to treatments. Microphthalmia-associated transcription factor (MITF) is the main factor that drives the melanoma phenotype switch and its expression correlates with each specific phenotype. Additionally, the expression of stemness-related factors has been associated with the invasive phenotype and goes along with an increase in aggressiveness and plasticity of the tumors.

**Figure 4 ijms-21-08274-f004:**
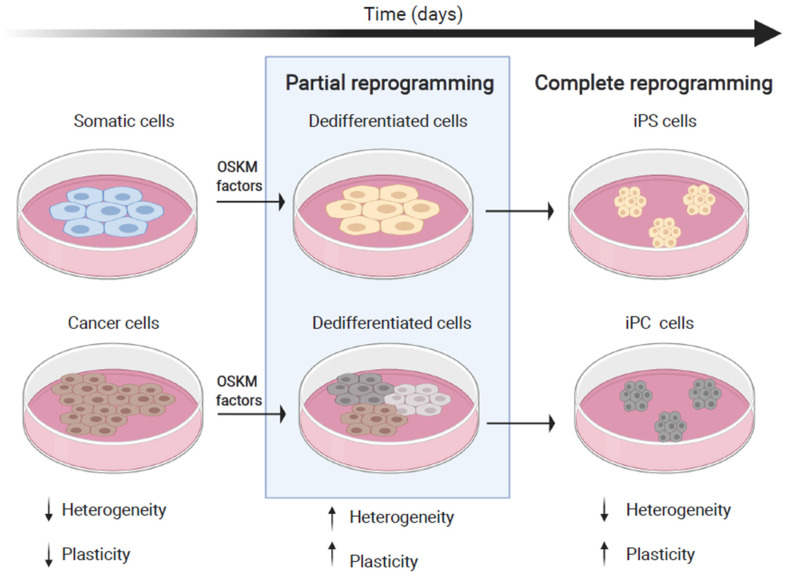
Partial reprogramming as a model for melanoma plasticity. Somatic and cancer cells can both be reprogrammed towards pluripotency, for example, by ectopically overexpressing the factors OSKM. Stopping the reprogramming process before completion, yields a population of partially reprogrammed, quite heterogeneous, dedifferentiated cells. During reprogramming, the plasticity of the cells increases gradually.

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
