# Peer review of "Cellular Reprogramming—A Model for Melanoma Cellular Plasticity"

_ijms, 2020, doi:10.3390/ijms21218274_

Round 1
Reviewer 1 Report
This is my opinion about the review: Cellular Reprogramming - a model for melanoma cellular plasticity.
The abstract is clear but repetitive. About topic of the article i think it may be useful to point out its importance in order to direct future studies aimed at understanding cellular plasticity of cancer cells and the use of complete and partial cellular reprogramming as a model to study melanoma cellular plasticity.
The development of the work is linear and smooth. I appreciate the easy reading of the text even if I notice some small errors (spaces, repetition of acronyms in full) that are easily identifiable with a thorough reading.
The presence of numerous figures enrich and enhance the work making it clear.
Author Response
Reviewer #1 (Comments to the Author):
This is my opinion about the review: Cellular Reprogramming - a model for melanoma cellular plasticity.
The abstract is clear but repetitive. About topic of the article i think it may be useful to point out its importance in order to direct future studies aimed at understanding cellular plasticity of cancer cells and the use of complete and partial cellular reprogramming as a model to study melanoma cellular plasticity.
The development of the work is linear and smooth. I appreciate the easy reading of the text even if I notice some small errors (spaces, repetition of acronyms in full) that are easily identifiable with a thorough reading.
The presence of numerous figures enrich and enhance the work making it clear
Response:
We really appreciate the suggestions and also the comments about our manuscript. Regarding some of the points mentioned before, we have made the following changes:
- We have eliminated a sentence in the abstract to make it less repetitive. (Lines 18-20 page 1).
- We also checked and corrected repetition of acronyms in full (Line 13 page 3; Line 36 page 9) and spaces (line 20 page 2; line 44 page 3; line 24 page 4; line 23 page 8).
3. We have included some ideas about the importance of using complete or partial reprogramming in future studies to understand melanoma cellular plasticity. (Lines 21-25 page 9).
Reviewer 2 Report
The Authors present an interesting work that shows the role of cellular reprogramming in cellular plasticity of melanoma. The topic offers a succinct and clear summary about how cellular reprogramming influences cancer progression and differentiated phenotype. Although the topic has been discussed in literature, the Authors treat the subject from a new perspective, which is interesting and useful to the scientific community. Overall, this is a very good study and well-supported. Only minor revisions are recommended.
Minor Revision
The Authors could add a section where deepen metabolic rewiring in melanoma and how metabolic rewiring may influence epigenetic machinery, drug effectiveness and resistance to standard treatment.
The Authors should mention:
- Ratnikov et al., Metabolic rewiring in melanoma. Oncogene. 2017 Jan 12;36(2):147-157. doi: 10.1038/onc.2016.198. Epub 2016 Jun 6.
Or other similar publications.
Author Response
Referee #2 (Comments to the Author):
The Authors present an interesting work that shows the role of cellular reprogramming in cellular plasticity of melanoma. The topic offers a succinct and clear summary about how cellular reprogramming influences cancer progression and differentiated phenotype. Although the topic has been discussed in literature, the Authors treat the subject from a new perspective, which is interesting and useful to the scientific community. Overall, this is a very good study and well-supported. Only minor revisions are recommended.
Minor Revision
Point 1: The Authors could add a section where deepen metabolic rewiring in melanoma and how metabolic rewiring may influence epigenetic machinery, drug effectiveness and resistance to standard treatment.
Response:
Thank you for the comment. We have extended this topic in the manuscript (Lines 1-10 page 7)
Point 2: The Authors should mention:
Ratnikov et al., Metabolic rewiring in melanoma. Oncogene. 2017 Jan 12;36(2):147-157. doi: 10.1038/onc.2016.198. Epub 2016 Jun 6.
Or other similar publications.
Response:
Thank you for this observation. We have added this reference and also some similar publications:
- Pavlova, N.N.; Thompson, C.B. The Emerging Hallmarks of Cancer Metabolism. Cell Metab. 2016, 23, 27–47, doi:10.1016/j.cmet.2015.12.006.
- Hill, K.S.; Kim, M. Decision to grow or to invade is at the flick of metabolic switch, PGC1α. Pigment Cell Melanoma Res. 2017, 30, 179–180, doi:10.1111/pcmr.12558.
- Ratnikov, B.I.; Scott, D.A.; Osterman, A.L.; Smith, J.W.; Ronai, Z.A. Metabolic rewiring in melanoma. Oncogene 2017, 36, 147–157, doi:10.1038/onc.2016.198.
- Bristot, I.J.; Kehl Dias, C.; Chapola, H.; Parsons, R.B.; Klamt, F. Metabolic rewiring in melanoma drug-resistant cells. Crit. Rev. Oncol. Hematol. 2020, 153, 102995, doi:10.1016/j.critrevonc.2020.102995.